# Particle-like topologies in light

Danica Sugic [1,2,3], Ramon Droop[4], Eileen Otte [4], Daniel Ehrmanntraut [4], Franco Nori [3,5], Janne Ruostekoski [6], Cornelia Denz[4] & Mark R. Dennis [1,2,7✉]

Three-dimensional (3D) topological states resemble truly localised, particle-like objects in physical space. Among the richest such structures are 3D skyrmions and hopfions, that realise integer topological numbers in their configuration via homotopic mappings from real space to the hypersphere (sphere in 4D space) or the 2D sphere. They have received tremendous attention as exotic textures in particle physics, cosmology, superfluids, and many other systems. Here we experimentally create and measure a topological 3D skyrmionic hopfion in fully structured light. By simultaneously tailoring the polarisation and phase profile, our beam establishes the skyrmionic mapping by realising every possible optical state in the propagation volume. The resulting light field's Stokes parameters and phase are synthesised into a Hopf fibration texture. We perform volumetric full-field reconstruction of the $\Pi_3$ mapping, measuring a quantised topological charge, or Skyrme number, of 0.945. Such topological state control opens avenues for 3D optical data encoding and metrology. The Hopf characterisation of the optical hypersphere endows a fresh perspective to topological optics, offering experimentally-accessible photonic analogues to the gamut of particle-like 3D topological textures, from condensed matter to high-energy physics.

---

[1] School of Physics and Astronomy, University of Birmingham, Birmingham B15 2TT, UK. [2] H H Wills Physics Laboratory, University of Bristol, Bristol BS8 1TL, UK. [3] Theoretical Quantum Physics Laboratory, RIKEN Cluster for Pioneering Research, Wako-shi, Saitama 351-0198, Japan. [4] Institute of Applied Physics and Center for Nonlinear Science (CeNoS), University of Muenster, 48149 Muenster, Germany. [5] Physics Department, University of Michigan, Ann Arbor, MI 48109-1040, USA. [6] Physics Department, Lancaster University, Lancaster LA1 4YB, UK. [7] EPSRC Centre for Doctoral Training in Topological Design, University of Birmingham, Birmingham B15 2TT, UK. ✉email: m.r.dennis@bham.ac.uk

Nontrivial 3D topology has inspired many descriptions of fundamental particles. Motivated by Lord Kelvin's knotted vortex atom hypothesis[1], Tony Skyrme[2] in 1961 proposed a topological model for nuclei: particle-like continuous fields in 3D space now called skyrmions. These map 3D real space to the hypersphere (i.e. the unit sphere in four dimensions, also known as the 3-sphere[3,4]), parametrising the field. The skyrmion configuration wraps around the hypersphere an integer number of times called the Skyrme number. Skyrmions are now seen as a particular example of more general 3D topological solitons[5–7], related to other topological textures such as monopoles and hopfions—the latter being fields with a 2-sphere parameter space (i.e. unit sphere in three dimensions). 3D topological textures have been studied theoretically as hypothetical objects in various systems, including high-energy physics[5,8], condensed matter[6,7,9,10], and early-universe cosmology[11]. In recent years, 3D skyrmions and hopfions have been experimentally realised in cold quantum matter[12,13] and liquid crystals[14].

So-called baby skyrmions are the two-dimensional (2D) counterpart of 3D skyrmions: fields in 2D physical space which map to, and wrap around, a 2-sphere parameter space. Their study is much more developed in theory and experiments, notably in non-singular superfluid vortices[15] including those imprinted by structured light[16], and especially magnetic systems[17]. Here the direction of spin at each point provides the 2-sphere parameter space, and magnetic skyrmion excitations have the potential to represent topological bits for low-power computer memory and processing[17]. Recently, 2D baby skyrmion configurations were created in optical systems, as the direction of electric field vectors, or photon spin, near a material interface[18,19], displaying dynamics similar to magnetic skyrmions[20]. In propagating laser light, optical polarisation can be structured into full Poincaré beams[21], which realise every state of elliptic polarisation in the transverse plane. These beams can also be interpreted as 2D baby skyrmions[22], since the Poincaré sphere, as the 2-sphere parameter space, parametrises transverse, elliptic polarisation states. However, 3D particle-like topological objects have not been considered either theoretically or experimentally in optical fields.

Optical realisations of 3D topological states can take various forms. Much interest has focused on singularity lines, such as optical vortices or polarisation singularities (e.g. C lines)[22]. In structured light, with amplitude, phase, and polarisation spatially varying, these can be woven into loops, links, and knots[23,24] and organise Möbius strips[25]. The state of elliptic polarisation is right- or left-handed circular (RH, LH) on C lines, often described as a skeleton of the complex optical polarisation field[26]. Topologically structured light has a wide range of applications including enhanced free-space optical communications[27] and advanced trapping[28], and is related to optical currents[29] and orbital angular momentum[30]. Singular lines are topologically characterised by the fundamental homotopy group $\Pi_1$. The homotopy group $\Pi_3$, on the other hand, defines topological particles such as 3D hopfions and skyrmions[5]. It is natural to ask whether these 3D excitations can be created in structured light.

Here we show the design, generation and measurement of a structured, propagating beam of laser light realising such a mapping, unifying particle-like 3D topologies in free-space optics with those studied in high-energy physics, cosmology and various kinds of condensed matter.

## Results

### The optical hypersphere of polarisation and phase.
Spatially extended polarised light is represented by a complex transverse electric field vector at each point **r** in the propagating beam. Its RH and LH components are represented by the complex-valued

scalar functions $E_R(\mathbf{r})$ and $E_L(\mathbf{r})$, and the pair $(E_R, E_L)$ which characterises the optical state at each point is assumed normalised, i.e.

$$(\text{Re}E_R)^2 + (\text{Im}E_R)^2 + (\text{Re}E_L)^2 + (\text{Im}E_L)^2 = 1. \tag{1}$$

Therefore, this normalised optical field defines a mapping from each point in 3D real space to a point on the 3-sphere, which we call the optical hypersphere. The optical hypersphere is conveniently parametrised using spinorial angles $\alpha, \beta, \gamma$:

$$E_R = \cos\frac{\beta}{2}e^{i(\gamma-\alpha)/2} \quad \text{and} \quad E_L = \sin\frac{\beta}{2}e^{i(\gamma+\alpha)/2}, \tag{2}$$

for $0 \leq \beta \leq \pi$, $-\pi < \alpha \leq \pi$ and $-2\pi < \gamma \leq 2\pi$. The angles $\alpha, \beta, \gamma$ have a direct interpretation in terms of the polarisation and phase of the electric field state: with $S_1, S_2, S_3$ the normalised Stokes parameters, $\alpha = \arctan(S_1, S_2)$ is the polarisation azimuth, and $\cos\beta = S_3$ is the polarisation ellipticity; $\gamma = \arg E_R + \arg E_L$ is the sum of the two electric field components' phases[26,31]. Further details of these parameters and their relationship with the hypersphere and the Poincaré sphere (2-sphere) parametrising polarisation may be found in Supplementary Note 1.

The full Poincaré sphere of polarisation states can be realised in a transverse plane of a structured light field, created from the superposition of two, differently structured, LH and RH beam components, similar to a full Poincaré beam[21]. At each spatial point, the optical field has some elliptical polarisation state characterised by $\alpha, \beta$. In 3D, points of constant elliptical polarisation lie on filaments, generalising RH and LH circular polarised C lines. 3D real space is filled by the set of polarisation filaments, constituting a polarisation texture (Fig. 1a). Each filament corresponds to a point on the Poincaré sphere (Fig. 1b), and many filaments cross each plane (Fig. 1c). Although the polarisation is fixed on the filaments, the optical phase smoothly varies along them (Fig. 1c, insets). Any 3D structured light field with varying transverse polarisation can be represented by such a texture.

The 3-sphere supports the Hopf fibration[4], a fibre bundle which divides it into linked circles. In the optical hypersphere, each fixed polarisation state (with $\alpha, \beta$ constant) traces out a circle as the phase $\gamma$ goes through a $4\pi$ cycle. The phase and polarisation parameters therefore realise the Hopf fibration in the optical hypersphere (this is explained in detail in Supplementary Note 1). The Poincaré sphere is interpreted here as the base space of the fibration[31]. We design a 3D structured beam that realises all the transverse states of light, including polarisation and phase, in its focal volume (real space). It displays the 3D Hopf fibration topology in a configuration we call a skyrmionic hopfion. The skyrmionic hopfion realises, in real space, an image of the Hopf fibration in the optical hypersphere. The fixed polarisation filaments can be represented as a 3D topological texture of entwined curves, in which each pair of loops are linked.

### Experimentally realising the skyrmionic hopfion.
We design the skyrmionic hopfion structure in light by superimposing carefully chosen combinations of vectorial Laguerre−Gauss beams[23,30] $LG_{\ell,p}$. The LH component, $E_L$, is chosen to be the Laguerre−Gauss beam $LG_{-1,0}$, with a negative-signed optical vortex along the beam axis[23]. The RH component, $E_R$, is chosen as a superposition of the Laguerre−Gauss beams $LG_{0,0}$ and $LG_{0,1}$, with a circular vortex loop in the focal plane centred on the axis[23]. Therefore, the net polarisation field has an RH C line along the axis, threading an LH C line loop in the focal plane. The C lines, at which $\beta = 0, \pi$, organise the rest of the texture: between them are nested tori with $\beta =$ constant, including the particular L surface of linear polarisation at $\beta = \pi/2$, analogous to vortices in

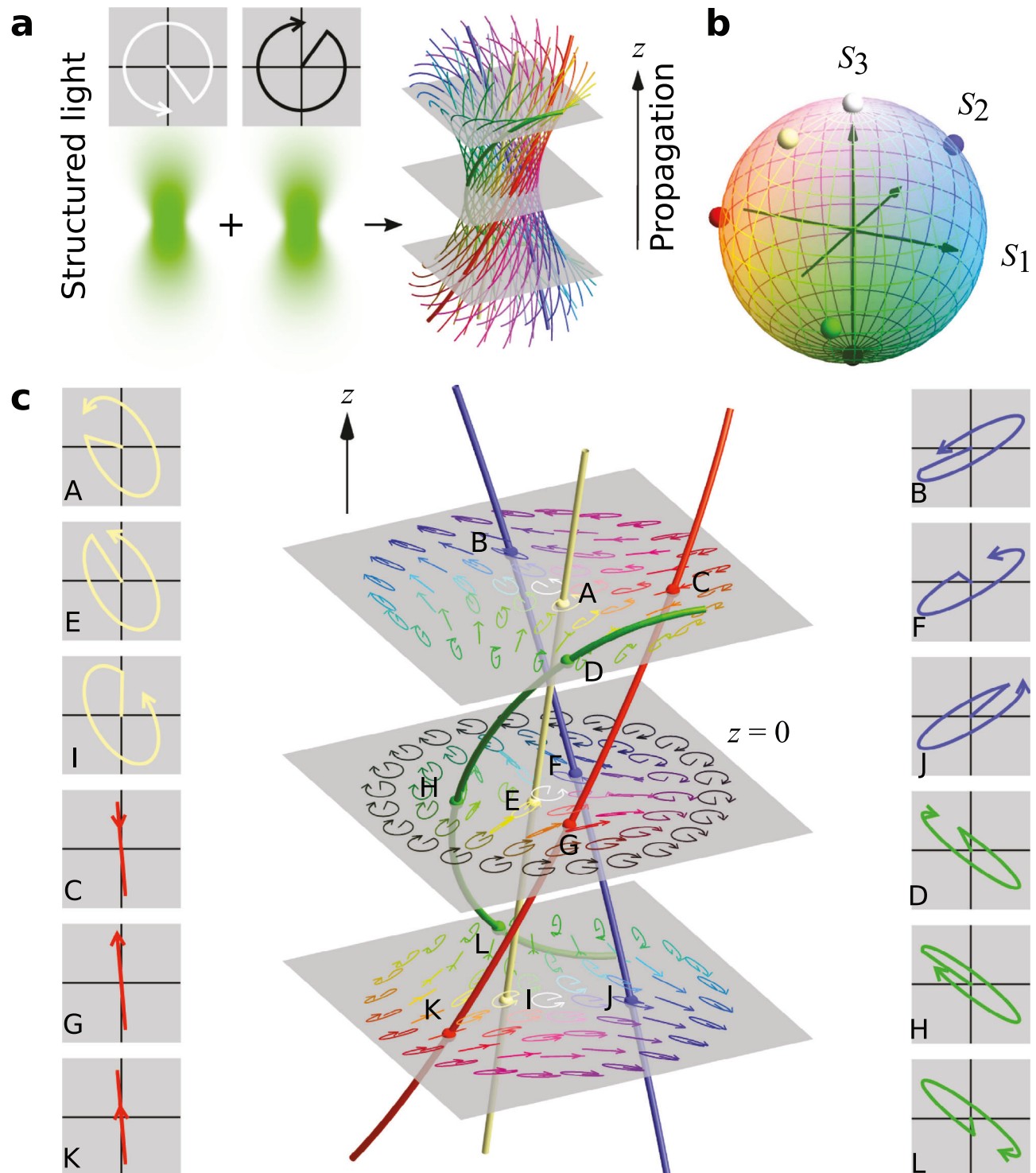

**Fig. 1 3D optical polarisation texture.** A light field with position-dependent transverse polarisation and phase is created in a volume from the superposition of RH and LH circularly polarised beams, whose amplitude and phases are carefully structured. Spatial points characterised by the same state of elliptic polarisation lie on the filaments (**a**). The 3D polarisation texture can be visualised by colouring the filaments according to the position of its polarisation ellipse on the Poincaré sphere (**b**). The azimuthal angle $\alpha$, representing the ellipse orientation, is coloured with the hues and the polar angle $\beta$, representing the ellipticity, is associated with the saturation levels. The sphere's poles, representing the circular polarised states, are black (LH) and white (RH). Each optical state also has a phase, represented by the position of the arrow along the polarisation ellipse. In the transverse plane in (**c**), states of light are fully described by colours and arrowed ellipses. Along filaments of constant polarisation, the phase on the ellipses varies smoothly, as shown in the insets for three representative planes.

other skyrmionic textures[9]. Details of the superposition optimisation are given in the "Methods" and Supplementary Note 2.

Experimentally, the RH and LH beam components are separately shaped by a spatial light modulator (SLM) (Fig. 2), before being combined in a joint beam path to shape the skyrmionic hopfion (Supplementary Fig. 4). The total polarisation state and phase of the resulting beam are measured at each point in the propagating volume via vectorial full-field reconstruction (VFFR, see "Methods"). For the VFFR, we combine established metrological techniques; namely, Stokes polarimetry, interferometry, and digital propagation[32]. Our approach explicitly relates measurements in different 2D planes, reconstructing the full 3D field volume. Further details of the experiment are given in the "Methods" and Supplementary Note 3.

The VFFR measurements reveal the polarisation Hopf fibration in the 3D light structure (Fig. 3a and Supplementary Video). The polarisation ellipticity is constant on nested tori, made up of polarisation filaments labelled by constant $\beta$ and varying azimuth $\alpha$ (Fig. 3b–d). Our polarimetric resolution identifies these filaments clearly, particularly the linking between pairs of loops. This resolution compares very well with experimentally measured hopfion structures in other systems, such as cold atoms[12,13] and liquid crystals[14]. As predicted (see Supplementary Notes 1 and 2), the two linked C lines (vortices in the superposed beams) are the topological skeleton of the hopfion structure, on which the rest of the polarisation texture hangs. They are not topologically privileged—all polarisation filaments are linked loops—but the C lines form the core filaments for the system of tori, including the L surface of linear polarisation. We anticipate C lines to play a similar structural role in other topological 3D polarisation textures.

Considering the shaped beams' phase as well as polarisation allows a comparison of the measured hopfion structure in real space (Fig. 4a, with phases along the shown filaments in Fig. 4b) with the optical hypersphere (Fig. 4c), parametrised by $\alpha, \beta, \gamma$. This direct comparison gives a volume-to-volume mapping (demonstrated by the grey cubes in Fig. 4a, c). The density of hypersphere volume with respect to real space volume is the topological Skyrme density $\Sigma$, which can be interpreted as a continuous measure of linking[33] of the polarisation filaments. Characteristic of 3D skyrmions[5,8,9], the real space integral of $\Sigma$, concentrated around the C line loop, integrates to a value very close to unity, covering the hypersphere of hypersolid angle $2\pi^2$, i.e. a Skyrme number of 1. The Skyrme number is the degree of the mapping from 3D real space to the hypersphere, corresponding to the element of the homotopy group $\Pi_3$. More details of this are provided in Supplementary Notes 1, 2 and 4.

Mathematically, the Skyrme density $\Sigma$ is the Jacobian determinant of the map from real space to the hypersphere (see Supplementary Note 4),

$$\Sigma = \frac{1}{16\pi^2} \nabla\gamma \cdot (\nabla\cos\beta \times \nabla\alpha). \quad (3)$$

This is the natural 3D generalisation of the 2D topological density for 2D skyrmions[13,15] (here, full Poincare beams), $\frac{1}{4\pi} \hat{\mathbf{z}} \cdot (\nabla\cos\beta \times \nabla\alpha)$. As the field parameters vary longitudinally as well as transversely, three parameters are needed to determine the full, continuous topological density determining the covering of the optical hypersphere, which is nonzero when the three gradient vectors are linearly independent. The topological density in Eq. (3) may be rewritten in terms of the normalised optical orbital current[29] $\mathbf{J}_o = \text{Im}[E_R^* \nabla E_R + E_L^* \nabla E_L]$,

$$\Sigma = \frac{1}{4\pi^2} \mathbf{J}_o \cdot \nabla \times \mathbf{J}_o. \quad (4)$$

Details are given in Supplementary Note 4. An analogous expression applies to 3D skyrmions in other systems[6,7], with an appropriate current or velocity substituted. It is also the topological helicity, describing knotted fields in high-energy physics[5], superfluids[6,7], magnetic fields and hydrodynamics[34]. Its appearance in Eq. (4) suggests a relation between the 3D Skyrme density of a polarisation field and the Poynting vector of optical energy flow.

We determine the Skyrme density explicitly from the measured data, as shown in Fig. 4d. The sum over the measured voxels gives a Skyrme number of 0.945, which is less than unity since low intensities limit the measured volume boundary. The corresponding covering of the optical hypersphere, with the image of the real space measurement boundary, is represented in Supplementary Fig. 10. Rather than a smooth interpolation of the optical field measurements, this density is determined discretely from a simplicial cell complex of spherical tetrahedra in the optical hypersphere arising from the measured data points. Details of the technique and its implementation are in the "Methods" and Supplementary Note 5. The value of the Skyrme number of the theoretical field, with the same boundary, is 0.997, consistent with the experimental error.

## Discussion
We have demonstrated the experimental construction of a 3D skyrmionic hopfion in the polarisation and phase pattern of a propagating light beam. The Hopf fibration is realised in the natural polarisation parameters from Eq. (2), a mapping from 3D real space to the 3D optical hypersphere, generalising the Poincaré sphere naturally by including phase.

Our experiment and analysis manifest several topological ideas not commonly emphasised in optics. Firstly, optical polarisation fields in 3D can have topological textures, analogous to textures in condensed matter, high-energy physics, etc. This might lead to further insights and possibilities for topologically structured light and its applications. Secondly, as a parameter space for the full vectorial light field, the optical hypersphere goes beyond the standard Poincaré sphere. The usual approach requires a Pancharatnam−Berry phase[35,36] to be included later, ignoring the fact that the optical field parameters define a manifold as natural as the 3-sphere. It is intriguing to speculate whether the machinery of the Poincaré sphere analysis of polarisation and Jones calculus may be cast in the optical hypersphere.

The 3D polarisation Skyrme density $\Sigma$ in Eqs. (3) and (4) can be used as a tool to analyse optical vectorial full fields. $\Sigma$ is the continuous topological charge density representing the abstract optical hypersphere volume covered by each real space point. $\Sigma = 0$ when the gradients of ellipticity, phase, and azimuth are linearly dependent, typically occurring along surfaces in 3D. The relation between $\Sigma$ and the optical orbital current suggests a subtle interplay between the Poynting vector[29] and energy −momentum fluxes with optical hypersphere topology (explored further in Supplementary Notes 1 and 4).

A smooth polarisation texture is disrupted at point singularities in the polarisation field, such as saddle points in the parameters $\alpha, \beta, \gamma$. As previously observed[24] in the reconstruction of Seifert surfaces spanning knotted optical singularities, these points are experimentally hard to control and limit the effective reconstruction of textures of polarisation lines. They do not affect the Skyrme number, and such points will lie on the surfaces $\Sigma = 0$.

Our experiments and theory demonstrate some higher-dimensional topological invariances possible in structured light. This formulation and measurement of an optical $\Pi_3$ invariant will lead to robust topological design principles for 3D optical fields for free-space optics and nanophotonics. These skyrmionic structures generalise to fields with higher degree Skyrme numbers, involving more complex superposed beams including knots and links,

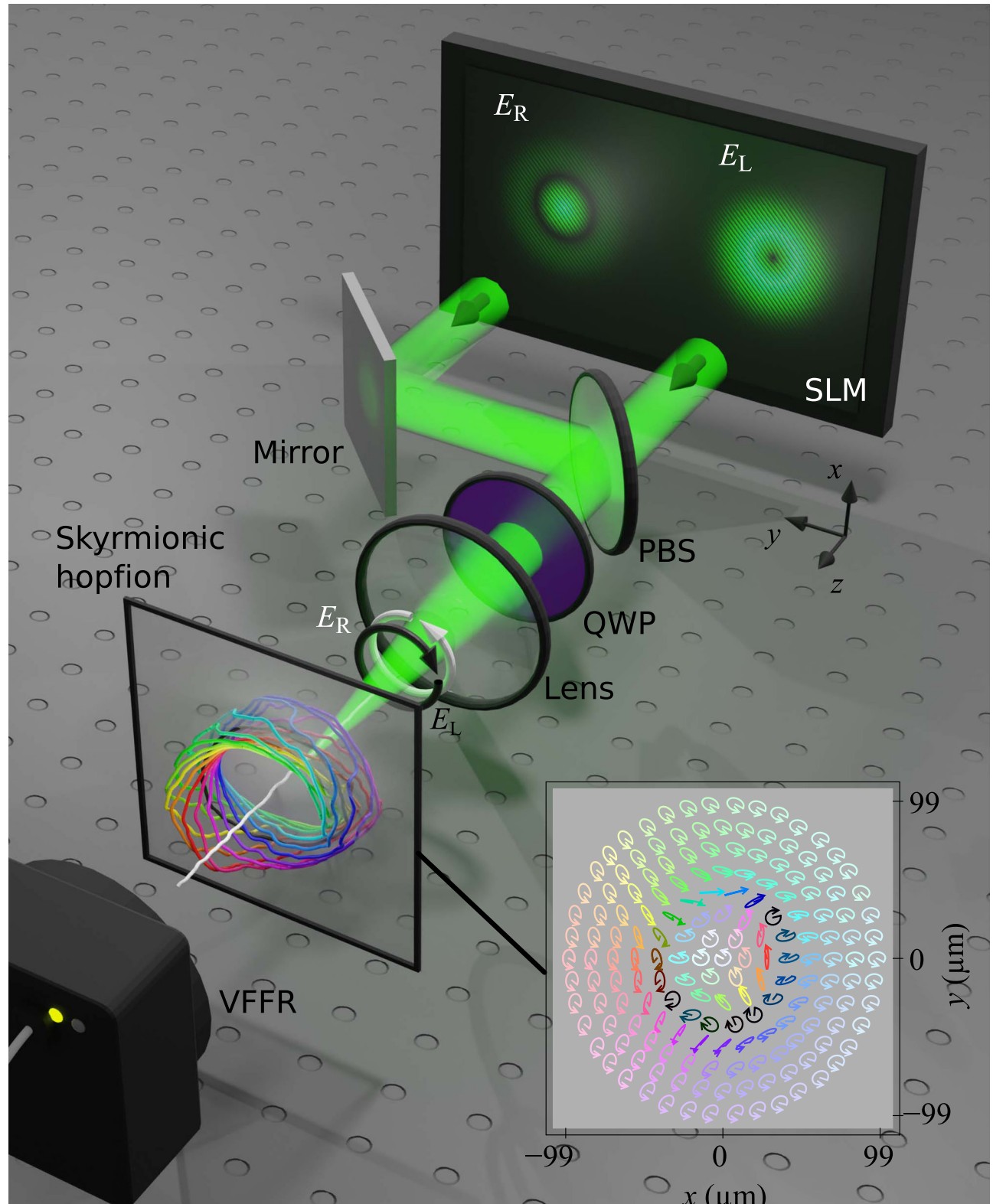

**Fig. 2 Sketch of experimental setup.** Beams $E_R$ and $E_L$ are generated on a spatial light modulator (SLM) and superimposed on-axis by a polarising beam splitter (PBS). A quarter-wave plate (QWP) transforms $E_L$ into left circular (black) and $E_R$ into right circular (white) polarisation. Around the focal spot of the lens, the skyrmionic hopfion appears in a cuboid of size 198.8 μm × 198.8 μm × 53.2 mm. The inset shows the polarisation texture in the focal plane (colour coded as the Poincaré sphere in Fig. 1b), consistent with the theory in Fig. 1c. Measurements of amplitude, phase and polarisation are enabled by volumetric full-field reconstruction (VFFR, see "Methods").

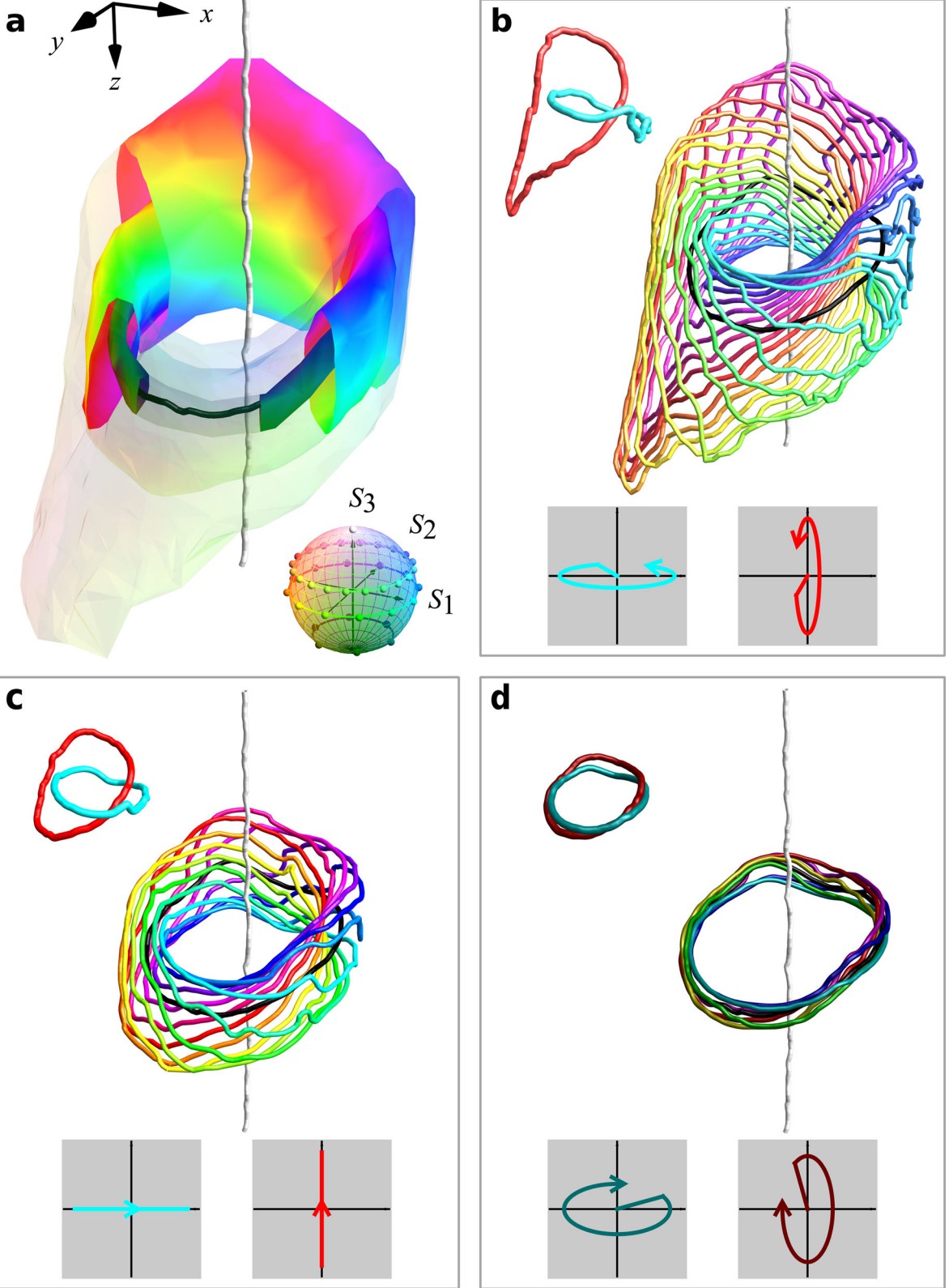

**Fig. 3 Visualising the topology of the focal volume.** The optical texture is reconstructed from the polarisation and phase measurements via the VFFR of the optical beam. The measured volume is coloured following the Poincaré sphere and reveals the topological structure of the Hopf fibration (**a**). Two C lines, the black loop and the threading straight white line, organise the texture into nested tori. Each toroidal surface represents points characterised by the same ellipticity. The colours wind nontrivially around each torus, and a few polarisation filaments making up these tori are shown in the insets: in (**b**), the lighter surface ($S_3 = 0.398$) is made of lines characterised by RH elliptic polarisation; in (**c**), the L surface ($S_3 = 0$) is made of lines along which the polarisation state is linear[23,26]; in (**d**), the darker surface ($S_3 = -0.775$) is made of lines characterised by LH elliptic polarisation. In each inset, the cyan and red filaments, corresponding to $\beta = 0, \pi$ are shown to form a Hopf link. Every pair of filaments in the texture link in this way, consistent with the Hopf fibration. The 3D rendering of this experimental skyrmionic hopfion is in the Supplementary Video.

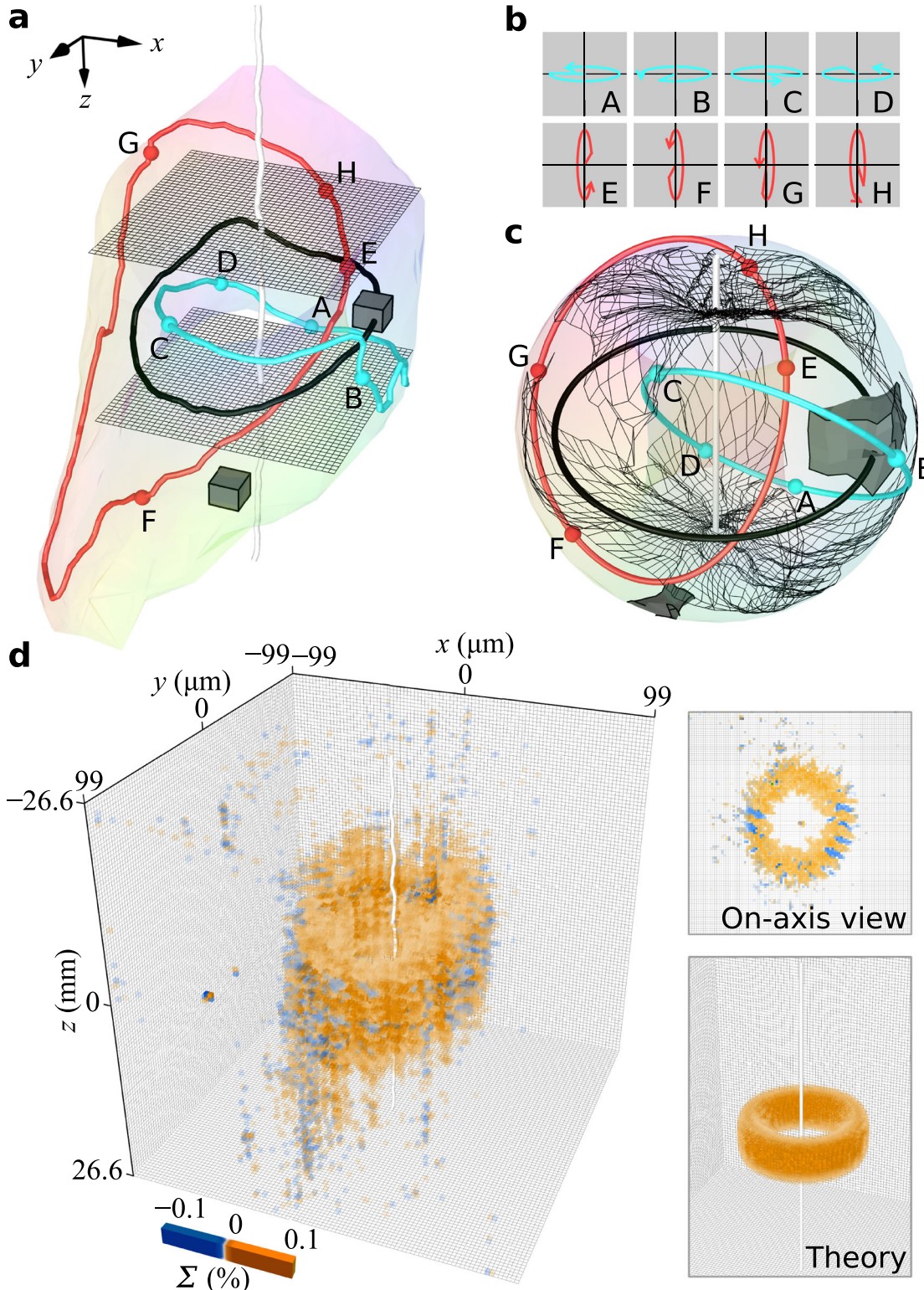

**Fig. 4 Measured Skyrme density and optical hypersphere.** The experimental polarisation hopfion is shown (**a**), with measured phases (**b**) around the two filaments shown ($\alpha = 0, \pi, S_3 = 0.398$, as in Fig. 3b). The hopfion in real space closely resembles the structure of the optical hypersphere parameter space in (**c**), shown in volume-preserving projection from the RH circular polarisation state at the focal point of real space. Several features make the nature of the topological mapping clear. The real-space filaments of constant polarisation are mapped to the smooth Hopf circles of fixed polarisation in the optical hypersphere. The images of two typical real space transverse planes (grey grids) are distorted in the parameter space. The cube in real space intersecting the LH C line (black loop) maps to a larger distorted cuboid, indicating a greater Skyrme density $\Sigma$ near this point. The cube away from the loop maps to a smaller cuboid, indicating a smaller $\Sigma$. In (**d**), the bounding cube represents the investigated focal volume in real space. The positive Skyrme density $\Sigma$ of our structured skyrmionic hopfion is concentrated around the LH C line with some positive and negative fluctuations visible around it. The upper inset shows the on-axis view (from $z = +\infty$) of the toroidal conformation. The measured Skyrme number, given by the sum of the cubes volume, is 0.942 (described in "Methods"). Theoretical predictions of the Skyrme density for the model field are shown in the lower inset, giving a Skyrme number of 0.997 (calculations in Supplementary Note 4).

offering a broader gamut of topological structures and integers that can be encoded in structured optical beams. This approach to topological beam shaping will offer further analogies with cold atoms, condensed matter and high-energy physics, offering the possibility of emulating, optically, exotic particle-like topologies from field theories not accessible otherwise in the laboratory.

## Methods

**Topological design of the optical skyrmionic hopfion**. The optical skyrmionic hopfion consists of the two scalar fields $E_R$ and $E_L$ representing the right- and left-handed field components respectively. These scalar components are appropriately structured to give the 3D topological texture described in the main text, effectively realising the topological mapping from 3D real space to the optical hypersphere. In these methods, we will refer to unnormalised field amplitudes $\psi_R$ and $\psi_L$ rather than their normalised counterparts: the beam intensity is $I = |\psi_R|^2 + |\psi_L|^2$, and $E_j = \psi_j/\sqrt{I}, j = R, L$.

The optical skyrmionic hopfion can be understood intuitively quite simply: the component $\psi_R$ should have a circular optical vortex line in the focal plane, concentric to the beam axis, and the component $\psi_L$ should have an optical vortex line along the beam axis. This realises all phases and polarisations (i.e., all points of the optical hypersphere), concentrated in a small propagation volume. These conditions can be realised by superpositions of Laguerre−Gauss (LG) modes[23,30]. The standard definition of these modes (given in Supplementary Note 2) defines $LG_{\ell,p}(R, \phi, z; w)$, depending on cylindrical coordinates in real space, $(R, \phi, z)$, with $\ell$ the azimuthal mode number, $p$ the radial mode number, and $w$ the waist width.

As discussed in Supplementary Note 4, the axial optical vortex line should have a negative sign, so we choose $\psi_L = 2cLG_{-1,0}(R, \phi, z; w)$, the simplest LG mode with an axial vortex of the correct sign, with $c$ a constant to be found and 2 included for calculational convenience. The vortex ring can be realised by the sum of two LG modes with $\ell=0$, $\psi_R = (-a + b)LG_{0,0}(R, \phi, z; w) - bLG_{0,1}(R, \phi, z; w)$, where $a$ and $b$ are parameters to be found. This guarantees the vortex ring to be in the focal plane $z = 0$, with a radius of $\sqrt{a/bw}$, provided $a, b > 0$. The coefficients $a$ and $b$ determine the intensity pattern around the vortex ring as well as its radius.

For fixed value of $w$ the optical skyrmionic hopfion is therefore realised for a range of values of $a, b$ and $c$. The different values of the parameters give very different shapes of the structure (residing in the polarisation parameters) and distribution of the overall intensity $I$. A preliminary exploration of these is given in Supplementary Note 4. The spreading nature of the gaussian beams means that it is not possible to cover the 3-sphere completely with polarisation states realised in 3-space. We therefore choose a superposition which maximises the volume of optical states in the optical hypersphere within the measured 3D volume in real space.

To be effectively generated and measured in the experiment, the values of the parameters are chosen to optimise the field configuration. To aid this optimisation, we introduce an extra scale size parameter $\mathscr{K}$, with $b = b_0\mathscr{K}^2$ and $c = c_0\mathscr{K}$. The 3D size of the skyrmionic hopfion scales according to $\mathscr{K}$, where now the vortex ring radius is $R_0 = \sqrt{a/b}(w/\mathscr{K})$. The remaining parameters $a, b_0, c_0$, determine the particle-like field distribution's shape. The parameters $a, b_0, c_0$ and $\mathscr{K}$ were chosen to ensure the experimental skyrmionic hopfion to be localised within the measured volume, in practice a cartesian cuboid centred around the focal point. We optimised against the criteria in the following list. (i) Vortex ring radius $R_0$ not larger than beam waist $w$ (this principle is also used in the design of optical vortex knots[37,38]). (ii) Concentrate intensity inside the measured volume, with $I \approx 0$ outside the measured volume. It was especially important to localise the intensity within the transverse cross-section, so as not to lose critical polarisation information. (iii) Distribute the intensity as evenly as possible within the measured volume. To maximise the quality of the measured polarisations we avoided regions of low intensity as much as possible where the polarisation state changes rapidly. (iv) Concentrate the Skyrme density (continuous topological charge density) within the measured volume, i.e. $\Sigma = 0$ outside the measured volume. The density $\Sigma$ is given in main text (Eqs. 3, 4), and described in detail in Supplementary Note 4. This enables a measured value of the Skyrme number very close to 1, as described in the remainder of the "Methods".

We proceeded by making an estimate of the parameters based on the topological 3D plots of the numerical models, and then improved these based on the quality of the experimental measurements.

The experimental setup, as described below, requires the Fourier transform of the beam superposition to be realised on the SLM, and the desired field is mathematically back-propagated through the paraxial lens system using Fourier optics[39]. The LG distributions when $z = 0$ are eigenfunctions of the Fourier transform operation. Thus, the real space LG mode $LG_{\ell,p}(R, \phi, z; w)$ corresponds, in Fourier space, to the 2D amplitude $i^{2p-|\ell|}LG^{2D}_{\ell,p}(\mathbf{q}_\perp; w_{\mathscr{F}})$, where $w_{\mathscr{F}}$ is the corresponding waist in the Fourier plane with transverse position $\mathbf{q}_\perp$. The holograms correspond to $-2ic_0\mathscr{K}LG^{2D}_{-1,0}(\mathbf{q}_\perp; w_{\mathscr{F}})$ for $\psi_L$ and $(-a + b_0\mathscr{K}^2)LG^{2D}_{0,0}(\mathbf{q}_\perp; w_{\mathscr{F}}) + b_0\mathscr{K}^2LG^{2D}_{0,1}(\mathbf{q}_\perp; w_{\mathscr{F}})$ for $\psi_R$. The coefficients do not depend on the Fourier waist $w_{\mathscr{F}}$, so the overall beam in real space scales linearly in radius $R$ and quadratically in propagation distance $z$ as $w_{\mathscr{F}}$ is varied. This quantity is chosen so that the skyrmionic hopfion has the desired size in real space whilst fully utilising the SLM.

In our optical system, $\lambda = 532$nm, the waist of the beam on the SLM is $w_{\mathscr{F}} = 6.252 \times 10^{-4}$m, and the imaging system given by lenses L1 and L2 (Supplementary Fig. 4b) halves the size of the beam. The resulting waist width is $w = 54.2\,\mu$m (giving a Rayleigh range $z_R = 34.7$mm). The measured volume is a cuboid, $|x| \leq x_{max}$, $|y| \leq y_{max}$, $|z| \leq z_{max}$, with $x_{max} = 3.13w = 170\,\mu$m; $y_{max} = 3.91w = 212\,\mu$m; $z_{max} = 0.768z_R = 26.6$mm. The values for the beam parameters were optimised in this range to be $a = 3$, $b_0 = 1.5$, $c_0 = 0.16$, $\mathscr{K} = 2.5$. In terms of the original parameters, this gives the values $a = 3$, $b = 9.4$, $c = 0.4$. With these choices, the LH C line ring is at $R_0 = 0.57w = 30.6\mu$m. The field configuration of this model field near the C line ring is shown in Supplementary Fig. 2b, resembling the corresponding Hopf fibration configuration (e.g., Supplementary Fig. 2a) closely.

**Optical system design**. The experimental skyrmionic hopfion field is the superposition of two structured beams of orthogonal circular polarisation, $\psi_R$ and $\psi_L$. Experimentally, these two scalar components are shaped by the amplitude and phase modulation of a collimated laser beam (horizontal linear polarisation, expanded) performed by a reflective phase-only SLM (Holoeye Pluto phase-only, $1920 \times 1080$ px HD display), shown in Supplementary Fig. 4b. The SLM is used in split-screen mode[40–42], with each half embedding the amplitude and phase information of $\psi_R$ and $\psi_L$ respectively. To optimise the beam quality, the Fourier hologram for each polarisation component is a $600 \times 600$ pixels square. This resolution was proven to produce all details of the transverse beam structure in the focal volume. The two holograms are placed so each receives approximately homogeneous illumination of the expanded input laser beam without losing too much intensity. The phase-only hologram is shown in Supplementary Fig. 4a.

To allow for amplitude modulation by a pure phase hologram, a weighted blazed grating is applied[43]. The desired scalar modes appear in the first diffraction order, which is spatially filtered by an aperture A in the conjugate plane of the SLM, generated by lens L1 (shown in Supplementary Fig. 4b). Fourier holograms are applied on the SLM, so that the desired beams are sculpted in the focus of the Fourier lens (L1), i.e. in the conjugate plane of the SLM. The hologram for each beam is normalised separately, taking advantage of the full modulation depth of the SLM for each beam individually.

The two beams are subsequently combined on-axis by an interferometric system. Before they are combined, the two beams are given orthogonal linear polarisations by a combination of a half wave plate (HWP) and a polarising beam splitter (PBS), allowing also for the adjustment of the beams' intensity ratio. This is a critical step to realise the complex polarisation structure: the HWP angle directly affects the relative strength of the two components and hence the coefficient $c$ in the field design described above.

After the beams are combined, a quarter-wave plate (QWP) transforms the orthogonal linear polarisation states into orthogonal circular polarisations. The imaging system given by lens L1 and L2 (Supplementary Fig. 4b) halves the size of the beam and L3 performs the final Fourier transform that gives the skyrmionic hopfion in its focal volume. The focal structure is magnified by lens L4 (×16) onto a CMOS camera (Cam; uEye SE (UI-1240SE), $1280 \times 1024$ px).

**Volumetric full-field reconstruction**. We retrieve the full-field information (transverse components of the paraxial beam) by reconstructing the polarisation and phase in the focal volume. Supplementary Fig. 5 shows five transverse planes at different positions in the propagation direction for the normalised Stokes parameters $S_1, S_2, S_3$ and the phases $\chi_R$ and $\chi_L$ of the RH and LH field components. The measurements in multiple transverse planes are performed via digital propagation[32] (see Supplementary Note 3). A detailed description of polarimetry[44] (Supplementary Fig. 4c) and transverse phase interferometry[45] (Supplementary Fig. 4d) can be found in Supplementary Note 3. The polarisation measurements across different planes are unaffected by the harmonic time dependence of the optical field and are directly stored into 3D arrays. However, when stacking volumetric phase measurements, the relative phase between neighbouring planes must be retrieved. First, we describe our procedure for connecting the transverse phase measurements to their neighbouring planes, and then we present our routine to minimise the experimental error in retrieving the field components.

The measured transverse phase structure per plane is constituted of the light field's propagation term, $e^{ikz}$ times the superposed LG structure described in the subsection "Topological design of the optical skyrmionic hopfion" of the "Methods" section above. This includes a Gouy phase factor $e^{-i\chi_G}$, where $\chi^G(z/z_R)$ is the $z$-dependent Gouy Phase, and a phase term varying radially and longitudinally (full Laguerre−Gauss modes equation is given in Supplementary Note 2), and a time-dependent phase offset due to the time varying phase relation between the measured and reference beams. In order to concentrate on the transverse variation, we circumvent the effect of $e^{ikz}$ within the measurements per

$z$-plane, thereby avoiding the effects of undersampling the electric field oscillation, by setting the distance between two transverse planes to a multiple of the wavelength ($100\lambda$), so the propagation factor $e^{ikz}$ is negligible. Next, we choose a transverse reference point ($\mathbf{r}_{\perp\mathrm{ref}}, z$) close to the optical axis ($R \approx 0$), so that the phase at this point is only affected by the $z$-dependent Gouy phase term of the LG beams and is unaffected by the other spatially varying phase factors. For each plane, the phase of the reference point is set to the same value, so the Gouy phase and the time-dependent phase offset are subtracted. In order to finalise the missing relation between different $z$-planes, the theoretical Gouy phase term $\chi^{\mathrm{G}}$ is added. Note that the Gouy phase represents an offset value per plane, only depending on the $z$-position but without any dependence on the transverse coordinates. Thus, the measurements themselves are not affected by this approach and, as a result, we correct for the errors in $z$ caused by the time-dependent variations in the measurement system. Supplementary Fig. 6 shows the $x = 0$ plane (longitudinal cut) of the theoretically expected (left) and the reconstructed (right) 3D phase structures of $\chi_{\mathrm{R}}$ and $\chi_{\mathrm{L}}$. This figure demonstrates that the reconstructed 3D phase distributions are consistent with the theoretical predictions.

Due to experimental errors (see Supplementary Note 3), the singularities of the differences of the phase of the two field components $\chi_{\mathrm{L}} - \chi_{\mathrm{R}}$ (wrapped between $-\pi$ and $\pi$) do not coincide with those of the polarimetrically-determined $\arctan(S_1, S_2)$ as the polarisation and phase measurement are independent. Observations of the 3D structure of the C lines from the polarisation measurements and the phase singularities from the phase measurements allow the systematic error to be minimised by shifting the polarisation measurements until the C line loop coincides with the singular loop of $\chi_{\mathrm{R}}$. Moreover, the overall error is reduced by redefining the Stokes parameters $S_1$ and $S_2$ as follows: $S_1 = 2\sqrt{s_0^2 - s_3^2}\cos(\chi_{\mathrm{L}} - \chi_{\mathrm{R}})/s_0$ and $S_2 = 2\sqrt{s_0^2 - s_3^2}\sin(\chi_{\mathrm{L}} - \chi_{\mathrm{R}})/s_0$. To finalise the volumetric full-field reconstruction we calculate the real and imaginary parts of the beam components from $E_{\mathrm{R}} = \sqrt{(s_0 + s_3)/2}\,e^{i\chi_{\mathrm{R}}}$, and $E_{\mathrm{L}} = \sqrt{(s_0 - s_3)/2}\,e^{i\chi_{\mathrm{L}}}$. The full field is used to calculate the Skyrme density of the optical field as described in the next subsection.

**Numerical calculation of experimental Skyrme number**. We measure the Skyrme number of the optical skyrmionic hopfion directly from the discretely sampled, measured data by taking advantage of the robustness of topology. This optimises the computational speed necessary to evaluate the Skyrme number from experimental measurements. The measured polarisation and phase at each point in real 3D space correspond to a point in the optical hypersphere. The 3D cubic lattice of measured voxels is mapped into a topology-preserving but distorted lattice in the optical hypersphere. An example for the ideal skyrmionics hopfion field (see Supplementary Note 4), is shown in Supplementary Fig. 9a, b. The measured Skyrme number is therefore based on this piecewise-linear mapping generated from the measured data points without interpolation. This approach can readily be used for measurements of other physical Skyrme-like maps, including lower dimensional ones (e.g. via triangular meshes).

The fully resolved experimental data in the focal volume give real space voxels forming a cuboidal grid. We are interested in the Skyrme density of the real space volume given by a cuboid with transverse size $\pm L_{\perp} = \pm 1.84\,w = \pm 99.4\,\mu\mathrm{m}$ and longitudinal size $\pm L_{\parallel} = \pm 0.768\,z_{\mathrm{R}} = \pm 26.6\,\mathrm{mm}$ as defined in Supplementary Note 4. Since the image of the cuboidal mesh covers the volume of the hypersphere, reducing the resolution maintains this filling. The numerical routine is made more time efficient by reducing the resolution to a cubic mesh of dimension $101 \times 101 \times 101$ in physical space, centred at the focal point. The voxels are centred at points labelled by $(i, j, k)$ with $1 \le i, j, k \le 100$. Each such point corresponds to a normalised 4D vector $\vec{n} = (\mathrm{Re}E_{\mathrm{R}}, \mathrm{Im}E_{\mathrm{R}}, \mathrm{Re}E_{\mathrm{L}}, \mathrm{Im}E_{\mathrm{L}})$ found via the VFFR method, giving a distorted cubic 3D grid in the optical hypersphere whose vertices are the points $\vec{n}(i, j, k)$. The distortion of the experimentally measured field is significantly greater than the example in Supplementary Fig. 9a, b, as can be seen in the images of the two real space planes in the optical hypersphere in main text (Fig. 4). Each elementary cube $C = C_{i,j,k}$ is labelled by $i, j, k$, with vertices $\vec{n}(i, j, k)$, $\vec{n}(i+1, j, k)$, $\vec{n}(i, j+1, k)$,... denoted by $c1$, ..., $c8$ as indicated in Supplementary Fig. 9c, d. The cube $C$ occupies a volume $\mathrm{Vol}(C)$ within the optical hypersphere. We numerically determine $\mathrm{Vol}(C)$ as follows.

An elementary topological cell in 3D is a tetrahedron (i.e. a 3-simplex[46], in the language of simplicial topology). We convert our cubic $i, j, k$ lattice into a 3D simplicial complex by decomposing each cube into five irregular tetrahedra. The resulting mesh of tetrahedra, where neighbours share triangular faces, edges and vertices, make up a 3D cell complex[46]. The tetrahedra can share the cube's vertices in two distinct ways, which are given by the following ordered sets of four vertices (see Supplementary Fig. 9c, d): (A), $(c1, c2, c4, c5)$, $(c4, c5, c7, c8)$, $(c5, c2, c7, c6)$, $(c2, c7, c3, c4)$, $(c5, c7, c2, c4)$ and (B), $(c1, c2, c3, c6)$, $(c3, c4, c1, c8)$, $(c5, c6, c1, c8)$, $(c7, c8, c3, c6)$, $(c8, c1, c3, c6)$. For any cubic lattice, cubes can be decomposed into two choices of tetrahedral mesh: cubes of type A at positions where the quantity $i + j + k$ is even (odd) and cubes of type B where $i + j + k$ is odd (even). We compute both types of 3D cell complex as a check of numerical accuracy. As a result, the measurement points in real space and measured values in the hypersphere define a piecewise-linear map representing the physical field.

In the hypersphere, the tetrahedra are constructed so the edges joining the vertices are geodesics. Each tetrahedron's four faces are spherical triangles, and along edges, pairs of faces meet at the dihedral angles $0 < \varphi_j < \pi$, for $j = 1, 2, \ldots, 6$. The formula for the 3D volume $\mathrm{Vol}(T)$ of an irregular spherical tetrahedron $T$ constructed in this way can be written explicitly in terms of dihedral angles by means of Murakami's formula[47] (see Supplementary Note 5). The contribution to the Skyrme number comes from the signed volumes $\mathrm{sign}\left[\det(\vec{n}_a, \vec{n}_b, \vec{n}_c, \vec{n}_d)\right]\mathrm{Vol}(T)$, where $\vec{n}_\ell$ with $\ell = \{a, b, c, d\}$ are 4D unit vectors pointing to the four vertices of a spherical tetrahedron $T$. Only tetrahedral cells included within a 3-dimensional hemisphere, whose volume is less than $\pi^2$, are considered. The sign of the volume comes from the ordering of the vertices with respect to the right-hand rule, where the triangular base $a, b, c$ follows the fingers and the vertex $d$ follows the thumb. When the volume is negative, the order of the vertices in real space and that of the vertices of the tetrahedron in the optical hypersphere are inverted. This follows the standard orientation rules of a 3-simplex.

At higher resolutions of the cubic lattice, the spherical tetrahedra are smaller, and the curved edges tend to become linear, and the spherical distortion can be neglected: the tetrahedron volume is better approximated by its flat-space analogue. The $101 \times 101 \times 101$ mesh defined above is consistent with the volume of the tetrahedra being within the range allowed by numerical precision.

The hyperspherical volume of the cube $C$ corresponds to the union of the volumes of the associated, neighbouring tetrahedra comprising $C$, and its volume $\mathrm{Vol}(C)$ is the sum of the signed spherical tetrahedra volumes $\mathrm{Vol}(T)$. The results for each such $\mathrm{Vol}(C_{i,j,k})$ are stored in two 3D arrays, one for each type of 3D cell complex. The experimental Skyrme number is found by adding together the volumes of all the hypersphere cubes with appropriate sign, normalised by the 3-sphere volume: $\sum_{i,j,k}\mathrm{Vol}(C_{i,j,k})/(2\pi^2)$ (see Supplementary Note 5). The measured 3D Skyrme number corresponds to the fraction of the hypersphere volume by the image of the measured volume of real space. The sums over all the elements in the arrays give Skyrme numbers 0.94521 and 0.94528 for the two kinds of mesh. We take the experimental Skyrme number to be the mean of these two numbers, 0.94524. It is straightforward to implement the vector calculation described here in a numerical algorithm in MATLAB or Python. The volumes of the cubes in the meshes can be calculated in parallel via high performance computers.

## Data availability
The experimental data are available from the corresponding author upon reasonable request.

## Code availability
The code for the Skyrme number calculation is available from the corresponding author upon reasonable request.

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

## Acknowledgements
We are grateful to Miguel Alonso, Michael Berry, Jörg Götte, Michael Morgan, Renzo Ricca, Paul Sutcliffe, Benny Jung-Shen Tai, Alexander Taylor, Teuntje Tijssen, Jonathan Watkins, Alessandro Zannotti and Shuang Zhang, and especially Benjamin Bode, David Foster and Ivan Smalyukh for conversations, advice, and research support. The numerical computations were performed using the University of Birmingham's BEAR Cloud service. D.S. and M.R.D. acknowledge financial support from the University of Birmingham, the Leverhulme Trust Research Programme RP2013-K-009 (SPOCK: Scientific Properties of Complex Knots) and the EPSRC Centre for Doctoral Training in Topological Design (EP/S02297X/1). R.D., D.E., E.O., and C.D. acknowledge partial support by the German Research Foundation (DFG), under project DE 486/22-1 and DE 486/23-1, as well as by the European Union (EU) Horizon 2020 programme, in the framework of the European Training Network ColOpt ITN 721465. J.R. acknowledges financial support from Engineering and Physical Sciences Research Council (EP/S002952/1 and EP/P026133/1). D.S. and F.N. are supported in part by: Nippon Telegraph and Telephone Corporation (NTT) Research, the Japan Science and Technology Agency (JST) [via the Quantum Leap Flagship Program (Q-LEAP), the Moonshot R&D Grant Number JPMJMS2061, and the Centers of Research Excellence in Science and Technology (CREST) Grant No. JPMJCR1676], the Japan Society for the Promotion of Science (JSPS) [via the Grants-in-Aid for Scientific Research (KAKENHI) Grant No. JP20H00134 and the JSPS–RFBR Grant No. JPJSBP120194828], the Army Research Office (ARO) (Grant No. W911NF-18-1-0358), the Asian Office of Aerospace Research and Development (AOARD) (via Grant No. FA2386-20-1-4069), and the Foundational Questions Institute Fund (FQXi) via Grant No. FQXi-IAF19-06.

## Author contributions
D.S. and R.D. equally share first authorship. D.S. and M.R.D. formulated the theory and developed the numerical methods with assistance from J.R.; R.D. and E.O. designed and performed the experiment, supported by D.E.; C.D., J.R. and M.R.D. provided explanations of data; F.N., C.D. and M.R.D. supervised the project. All authors have approved the submitted version.

## Competing interests
The authors declare no competing interests.
