## [Peer Review File · Nature Communications]

1 REVIEWERS' COMMENTS

2

3 Reviewer #1 (Remarks to the Author):

4

5 Qualitative features of vector fields on the plane and in three-dimensional space are an important
6 part of the general theory of oscillations and waves. For example, in structured light, with
7 amplitude, phase, and polarization spatially varying, these can be woven into loops, links and
8 knots, and organize Möbius strips. Characterization of the topological properties of optical vector
9 electromagnetic fields is an extremely difficult task. The article proposes an original idea and a
10 new tool for the analysis of such fields, based on the construction of a three-dimensional skyrmion
11 hopfion in polarization and phase diagram of a propagating light beam. The experiments and
12 theory presented in the article demonstrate new topological invariants that are possible in a
13 structured light. Overall, the paper is well written, and I recommend it for publication.

14

15

16 Reviewer #2 (Remarks to the Author):

17

18 Mark Dennis and coworkers report on creating 3D Hopfions of Skyrmions in 3D structured light.
19 This manuscript is very topical and exciting and the results are extremely impressiv. The topic fits
20 in the scope of Nature Communications and should be published.
21 The results are extremely clean and convincing, the paper is very logically structured and clearly
22 written, and the additional information and the renderings (and movies) are beautiful.

23

24 The paper should be published in Nat Comm as is.